# Biological Compositions of Canine Amniotic Membrane and Its Extracts and the Investigation of Corneal Wound Healing Efficacy In Vitro

**DOI:** 10.3390/vetsci9050227

**Published:** 2022-05-09

**Authors:** Chompunut Permkam, Gunnaporn Suriyaphol, Sujin Sirisawadi, Nalinee Tuntivanich

**Affiliations:** 1Veterinary Bioscience Program, Department of Veterinary Anatomy, Faculty of Veterinary Science, Chulalongkorn University, Bangkok 10330, Thailand; chompunut.p@student.chula.ac.th; 2Biochemistry Unit, Department of Veterinary Physiology, Faculty of Veterinary Science, Chulalongkorn University, Bangkok 10330, Thailand; gunnaporn.v@chula.ac.th (G.S.); sujin.s@chula.ac.th (S.S.); 3Department of Veterinary Surgery, Faculty of Veterinary Science, Chulalongkorn University, Bangkok 10330, Thailand

**Keywords:** amniotic membrane, canine, cornea, extract, lyophilized, wound healing

## Abstract

The usage of canine amniotic membrane (cAM) is mainly of interest in veterinary ophthalmology. Topical formulations of cAM could deliver the beneficial properties of cAM without the need for surgical intervention. The present study aimed to investigate biological compositions of cAM and its extracts, including their corneal wound healing efficacy. In this study, canine amniotic membrane extract (cAME) and lyophilized canine amniotic membrane extract (cAMX) were developed. Bioactive molecules related to corneal wound healing, including hepatocyte growth factor, tissue inhibitor of metalloproteinase-1 and -2, Thrombospondin-1 and Interleukin-1 receptor antagonist were studied at both gene and protein expression levels. Cell viability and wound healing assays were investigated for the possibility of cAME and cAMX as topical applications. The results demonstrated that all of the relevant genes and proteins were detected in cAM, cAME and cAMX. Both cAME and cAMX showed wound healing properties in vitro and cAME at 1.0 mg/mL concentration appeared to have the best healing efficacy. In conclusion, cAME and cAMX generated for topical use provided promising results in the healing of corneal defects.

## 1. Introduction

The amniotic membrane (AM) is the innermost layer of the placenta. Its structure consists of three different parts, which are a single layer of epithelial cells, a basement membrane and a stroma [1]. The AM gains its popularity as a surgical biomaterial because it provides a base for structural reestablishment and contains several bioactive molecules such as growth factors, cytokines and protease inhibitors, all of which have an effect on cell proliferation as well as anti-inflammation, anti-angiogenesis, anti-scarring and regeneration properties [2]. These AM properties play a pivotal role in repairing the ocular surface [3,4,5].

The use of AM requires surgical procedures and general anesthesia, which has a high risk and a high cost to operate. Therefore, several amniotic membrane derivatives, such as amniotic membrane extract (AME), amniotic membrane homogenate (AMH), amniotic membrane suspension (AMS) and amniotic membrane extract eye drop (AMEED) have been developed as topical formulations. The advantages of these formulations are that they deliver the beneficial components of AM to the application sites without the need for surgical manipulations and are also relatively inexpensive [6].

In the veterinary field, several studies have reported potential replacement of human amniotic membrane (hAM) in ocular surface reconstruction by using AM from other animal species, such as horses [7], cows [8], pigs [9] and dogs [10,11], which provided satisfying clinical outcomes. This study was focused on canine amniotic membrane (cAM) that was considered biological waste from the operation room and its availability in the animal hospital.

Canine amniotic membrane has been developed and clinically applied for many corneal reconstruction procedures in dogs, such as corneoscleral mass [10], corneal dermoids [11], deep corneal ulcers [12], complicated corneal ulcers [13], descemetocele and corneal perforation [14]. In recent years, cAM has become of interest as one of the most promising potential stem cell precursors in regenerative medicine [15,16,17]. Although the use of cAM has been widely accepted in clinical applications, basic knowledge regarding its biochemical components is very limited. Moreover, topical formulations of cAM such as cAM extract (cAME) and lyophilized cAM extract (cAMX) are interesting because they have potentially reduced surgical complications, are convenient to use and can be frequently prescribed as much as needed [6].

The investigations of specific bioactive molecules associated with the corneal wound healing process were performed at both gene and protein expression levels. Hepatocyte growth factor (HGF) plays an important role in the promotion of corneal epithelial cell (CEC) proliferation and suppression of ocular inflammation both in vitro and in vivo [18]. Tissue inhibitor of metalloproteinase-1 and -2 (TIMP-1 and TIMP-2) have anti-inflammatory and anti-angiogenesis properties by the inhibition of matrix metalloproteinase activity [19,20]. Thrombospondin-1 (TSP-1) is a potent angiogenic inhibitor which acts directly through its effects on endothelial cell proliferation, migration and apoptosis and by antagonizing the activity of vascular endothelial growth factor [21,22]. Interleukin-1 receptor antagonist (IL-1RA) can suppress the corneal inflammation reaction by reducing the influx of inflammatory cells into the cornea [23]. This study aimed to study the molecular characteristics of different preparations of cAM, cAME and cAMX. In addition, a corneal wound healing assay was also performed to investigate the feasibility of cAME and cAMX for clinical applications.

## 2. Materials and Methods

### 2.1. Canine Amniotic Membrane Collection and Preparation

One hundred and twenty placentas from healthy puppies were collected after cesarean sections in the operation room of Small Animal Teaching Hospital, Faculty of Veterinary Science, Chulalongkorn University, Bangkok, Thailand. All pregnant dogs aged 1–8 years were clinically healthy and had completed their vaccination program. Dogs with history of abortion, systemic inflammation or infection during pregnancy, or signs of inflammation of cAM, were excluded. Placental tissue and cAM in the experiments were in good condition. The ethical consideration of cAM collection was performed under the Institutional Animal Care and Use Committee of Chulalongkorn University (CU-IACUC) guidelines. After delivery, placentas were processed with sterile techniques in a laminar flow cabinet. Amnion was separated from chorion and the rest of the placenta, then washed with sterile phosphate buffer saline (PBS) containing 50 U/mL penicillin-streptomycin (Merck KGaA, Darmstadt, Germany), 0.1 mg/mL gentamicin (Merck KGaA, Darmstadt, Germany) and 2.5 µg/mL amphotericin B (Merck KGaA, Darmstadt, Germany). A portion of cAM with clear appearance, minimal to absent blood vessels, or absent any signs of inflammation was included in this study.

### 2.2. Canine Amniotic Membrane Preparation

A total number of 120 cAMs were used in this study. 84 cAMs were used for gene and protein expression studies, while the other 36 cAMs were used for cell viability and wound healing assays. Eighty-four cAMs were equally divided into 3 groups according to their preparation methods. Group 1, cAM (*n* = 28), was immersed in RNAlater™ solution (Thermo Fisher Scientific, Waltham, MA, USA) immediately after washing procedure, stored at 4 °C overnight, then kept at −20 °C until performing qRT-PCR and Western blot analysis. Group 2, cAM extract (cAME, *n* = 28), was pulverized using liquid nitrogen for 3 repetitions until it became fine particles. After thawing, the suspension was collected and centrifuged at 20,400× *g* for 15 min at 4 °C. The supernatant was collected and stored (ranging from 1–4 months) at −80 °C until the experiment. Group 3, lyophilized cAME (cAMX, *n* = 28), was prepared using a similar process to cAME. After the supernatant had been kept at −80 °C overnight, it was directly transferred to the freeze dry unit (Labconco LYPH·LOCK^®^4.5, Kansas City, MO, USA). The sample had been lyophilized at −70 °C, 2000-micron Hg vacuum for 8 h. As recommended by the manufacturer, the tubes containing cAME powder were kept at −20 °C until analysis (Figure 1). cAM, cAME and cAMX were then uniformly separated into 2 subgroups (*n* = 14 each) for further gene and protein expression studies.

### 2.3. Gene Expression Study by Quantitative Reverse Transcription Polymerase Chain Reaction (qRT-PCR)

#### 2.3.1. RNA Isolation

The total RNA of cAM, cAME and cAMX were extracted using a Nucleospin^®^RNA isolation kit (Macherey-Nagel GmbH & Co. KG, Düran, Germany). For cAM, RNALater™ solution was discarded. The membrane was washed with sterile PBS, patch-dried with sterile gauze and pulverized using liquid nitrogen. Thereafter, 600 µL of RNA isolation buffer was added to the tissue particles. In the case of cAME, the sample was thawed at room temperature and mixed with 600 µL of RNA isolation buffer. In the case of cAMX, the cAM powder was resuspended with 600 µL of RNA isolation buffer. RNA isolation was performed following the manufacturer’s protocol. Afterwards, genomic DNA was discarded using TURBO DNA-*free*™ kit (Thermo Fisher Scientific, Waltham, MA, USA). RNA concentration and purification were assessed using NanoDrop-1000 (Thermo Fisher Scientific, Waltham, MA, USA).

#### 2.3.2. Quantitative Reverse Transcription Polymerase Chain Reaction (qRT-PCR)

Complementary DNA (cDNA) was synthesized from DNase-treated RNA using the SuperScript™ III First-Strand Synthesis System for RT-PCR (Thermo Fisher Scientific, Waltham, MA, USA), according to the manufacturer’s recommendations. The cDNA was subjected to further qPCR reaction using the CFX Connect Real-Time PCR Detection System (Bio-Rad Laboratories Inc., Hercules, CA, USA) using the KAPA SYBR^®^ Fast qPCR Kit Mastermix Universal for detection (KAPA Biosystems, Wilmington, MA, USA). The corresponding set of primers (Table 1) were designed using Primer3 software version 4.0. The In-Silico PCR program was used to detect the virtual PCR opposing the canine reference genome (CanFam 3.1) and the Basic Local Alignment Search Tool (BLAST) was used to validate the specificity of each primer.

PCR amplifications were performed under the following conditions: pre-denaturation at 95 °C for 2 min, followed by 40 cycles of PCR reaction consisting of denaturation at 95 °C for 3 s, annealing at 60 °C for 20 s and extension at 72 °C for 1 s. Each sample was obtained in duplicate and PCR was performed in three independent runs.

#### 2.3.3. Statistical Analysis

Data from qRT-PCR were statistically analyzed using Relative Expression Software Tool (REST) 2009 (Qiagen GmBH, Hilden, Germany). Statistical significance was considered at *p* < 0.05.

### 2.4. Protein Expression Study by Western Blot Analysis

#### 2.4.1. Protein Extraction

For cAM, membrane was pulverized and resuspended with 0.5% sodium dodecyl sulfate (SDS), then centrifuged at 20,400× *g* for 15 min at 4 °C. The supernatants were collected. cAME was liquified at room temperature and 0.5% SDS was added and the mixture was vortexed. cAMX powder was resuspended in 0.5% SDS and the mixture was vortexed until cAMX had completely dissolved. The extracted proteins from all groups were measured for protein concentration using a Modified Lowry Protein Assay Kit (Thermo Fisher Scientific, Waltham, MA, USA). A protein concentration of 20 µg from each sample was aliquoted and stored at −20 °C for Western blot analysis.

#### 2.4.2. Western Blot Analysis

20 µg of protein samples were separated by 12% sodium dodecyl sulfate polyacrylamide gel electrophoresis (SDS-PAGE) at 90 volts for 90 min, then transferred onto a polyvinylidene difluoride (PVDF) membrane using a Trans-Blot Turbo Transfer System (Bio-Rad Laboratories Inc., Hercules, CA, USA). A Pierce™ Reversible Protein Stain Kit for PVDF Membranes (Thermo Fisher Scientific, Waltham, MA, USA) was used for total protein staining. Afterwards, non-specific proteins were blocked using 5% bovine serum albumin (Merck KGaA, Darmstadt, Germany) in Tris buffer saline and 0.1% Tween-20 (TBS-T) at room temperature for 1 h followed by incubation at 4 °C overnight with primary antibodies specific to the relevant proteins as follows: HGF and IL-1RA (1:500, Thermo Fisher Scientific, Waltham, MA, USA); TIMP-1 and TIMP-2 (1:1000, Thermo Fisher Scientific, Waltham, MA, USA); and TSP-1 (1:500, Santa Cruz Biotechnology, Dallas, TX, USA). On the next day, secondary antibodies of anti-rabbit IgG HRP conjugated antibody for HGF and IL-1RA (1:5000, R&D Systems Inc., Minneapolis, MN, USA) and m-IgGk BP-HRP for TIMP-1, TIMP-2 and TSP-1 (1:10,000, Santa Cruz Biotechnology, Dallas, TX, USA) were incubated at room temperature for 1 h, then Clarity™ Western ECL Substrate (Bio-Rad Laboratories Inc., Hercules, CA, USA) was used for protein band detection and the ChemiDoc Imaging System (Bio-Rad Laboratories Inc., Hercules, CA, USA) was used for visualization. Band intensities of the target proteins were analyzed using Bio-Rad Image Lab Software (version 6.0.1, Bio-Rad Laboratories Inc., Hercules, CA, USA). Western blot analysis of each protein was performed in triplicate.

#### 2.4.3. Statistical Analysis

Relative expressions of specific protein band intensities were normalized with total protein intensities. Comparison among groups was conducted using the Kruskal–Wallis test with Dunn’s multiple comparison using GraphPad Prism version 9.2.0 (GraphPad Software, San Diego, CA, USA) with a significance level at *p* < 0.05.

### 2.5. Investigation of Corneal Wound Healing Efficacy of cAME and cAMX

#### 2.5.1. Preparation of cAME and cAMX

Thirty-six cAM membranes were collected and processed as described in cAM collection. Each membrane was patch-dried with sterile gauzes before processing, then pulverized with liquid nitrogen. After thawing, the membrane solution was collected and centrifuged at 20,400× *g*, 4 °C for 30 min and its supernatant was transferred to 2 tubes. One was kept at −80 °C to serve immediately as cAME, while the other was kept at −80 °C overnight prior to the freeze-drying process to serve as cAMX. After freeze-drying, the membrane powder (cAMX) was kept at −20 °C until use.

Prior to the experiment, all tubes of cAME were thawed at room temperature. cAMX tubes were resuspended with 1 mL PBS, then both were filter sterilized using a 0.2-µm syringe filter (Merck KGaA, Darmstadt, Germany). cAME and cAMX (both 200 µL) were equally transferred to 2 new sterile tubes, one tube for protein concentration measurement by the Modified Lowry technique and the other tube for bacterial identification at the Veterinary Diagnostic Laboratory, Faculty of Veterinary Science, Chulalongkorn University, Bangkok, Thailand.

#### 2.5.2. Human Corneal Epithelial Cell Culture

The cryopreserved immortalized human corneal epithelial cells (hCECs, CRL-11516: ATCC, Manassas, VA, USA) were thawed and cultured in 75 cm^2^ tissue flasks containing complete growth media, including Gibco™ Keratinocyte-Serum Free Medium (KSFM) with 0.05 mg/mL bovine pituitary extract (BPE), 5 ng/mL epidermal growth factor (EGF) supplementation (Thermo Fisher Scientific, Waltham, MA, USA) and 0.005 mg/mL Gibco™ Insulin-Transferrin-Selenium Supplement 100× (Thermo Fisher Scientific, Waltham, MA, USA). The culture flasks were maintained at 37 °C and in 5% CO_2_. Culture media were changed every 2 consecutive days. When the cells had reached 80% confluency, subcultures were performed using 0.05% (*w*/*v*) Gibco™ Trypsin- 0.53mM EDTA (Thermo Fisher Scientific, Waltham, MA, USA). After discarding the Trypsin-EDTA solution, the cells were resuspended with fresh growth media.

#### 2.5.3. Cell Viability Assay

The hCECs were seeded at a concentration of 1 × 10^4^ cells/well into 96-well plates and incubated at 37 °C, in 5% CO_2_ until confluence. Thereafter, the media were removed and replaced with KSFM for cell starvation overnight. On the next day, the media were discarded and the experiment was performed using 100 µL of KSFM as a control and 100 µL of KSFM containing various concentration of cAME and cAMX (0.5, 1.0, 1.5, 2.0 mg/mL). The culture cells were incubated for 0, 24 and 48 h at 37 °C and in 5% CO_2_. After reaching each time point, media were discarded, 100 µL of KSFM containing 5 mg/mL MTT solution (Invitrogen, MA, USA) was added to each well and the cells were incubated for 4 h. After discarding the media containing MTT, 100 µL of the DMSO solution was added to each well and the cells were incubated for 30 min. Absorbance at 570 nm wavelength was assessed using a Multiskan Sky Microplate Spectrophotometer (Thermo Fisher Scientific, Waltham MA, USA). The reactions were analyzed in duplicate and the experiments were performed in triplicate.

#### 2.5.4. Statistical Analysis

Optical density data of hCECs cultured in different concentrations of cAME and cAMX were analyzed and percentages of cell viability were calculated. Comparison among concentrations was performed using two-way analysis of variance (ANOVA) with multiple comparison by GraphPad Prism version 9.2.0 (GraphPad Software, San Diego, CA, USA). The two most appropriate cAME and cAMX concentrations were selected for the wound-healing study.

#### 2.5.5. Wound Healing Assay

A confluent monolayer of hCECs was obtained from seeding 2 × 10^5^ cells per well in 6-well plates and incubating at 37 °C in 5% CO_2_ for 48 h. The media were discarded and replaced with KSFM for cell starvation overnight. On the next day, cells were wounded by manually cross scraping with a sterile 200 µL pipette tip and washed twice with sterile PBS to discard cell residues. hCECs were incubated with complete growth media supplemented with 0.5 and 1 mg/mL concentration of cAME or cAMX. Medium containing 5% fetal bovine serum (FBS) was used as the positive control, while medium without supplementation was used as the negative control. Each concentration was obtained in duplicate and each experiment was assessed in triplicate.

The original wound sizes were microscopically photographed and wound areas were calculated using ImageJ Software (version 1.52a, NIH, Bethesda, MD, USA). Wound closures were investigated under a light microscope 24, 48 and 72 h after wounding. Wound edges were assessed from photographs. Free-hand tracing of wound margin was used to assess wound areas at the beginning and at 24-, 48- and 72-h post-wounding. Thereafter, the percentage of wound closure was calculated using the following equation:$$\mathrm{Wound}\;\mathrm{closure}\;\left(\%\right)=\frac{\mathrm{wound}\;\mathrm{area}\;\mathrm{at}\;\mathrm{observation}\;\mathrm{time}-\mathrm{wound}\;\mathrm{area}\;\mathrm{at}\;\mathrm{the}\;\mathrm{beginning}}{\mathrm{wound}\;\mathrm{area}\;\mathrm{at}\;\mathrm{the}\;\mathrm{beginning}}\;\times \;100$$

#### 2.5.6. Statistical Analysis

The percentages of wound closure from cAME and cAMX treatment were calculated. Data are presented as percentage ± standard error of mean (SEM). Repeated measuring of two-way analysis of variance (ANOVA) with multiple comparison using GraphPad Prism version 9.2.0 (GraphPad Software, San Diego, CA, USA) was performed for statistical analysis. The significance level was considered at *p* < 0.05.

## 3. Results

### 3.1. cAME and cAMX Can Be Generated from cAM

Canine amniotic membrane had an average weight of 0.98 g per piece (range 0.61–1.39 g) (Figure 2A). cAME had the characteristics of a transparent-yellowish, viscous solution (Figure 2B). The protein concentration was 3.075 ± 0.102 mg/mL on average (range 2.469–3.650 mg/mL). cAMX had the character of a creamy to whitish fine powder (Figure 2C). cAMX had a weight of approximately 39.04 mg per tube (range 7.73–91.73 mg) after the lyophilization process. After resuspension with PBS and vortexing until absolutely dissolved, cAMX had similar characters to cAME and its average protein concentration was 2.541 ± 0.232 mg/mL (range 1.000–3.803 mg/mL).

### 3.2. Biological Compositions Associated with Corneal Wound Healing Were Identified in cAM, cAME and cAMX Both Gene and Protein Expression Levels

All of the relevant genes were detected in cAM, cAME and cAMX. Significant upregulation of TIMP-1 and IL-1RA genes was observed in both cAME and cAMX when compared to the cAM control. Conversely, expression of HGF and TSP-1 genes was significantly lowered in cAMX compared to cAM. When comparing cAMX to cAME, significant downregulation of HGF and IL-1RA genes was found (Table 2).

Western blot analysis revealed the detection of all relevant proteins in cAM, cAME and cAMX. cAM showed the highest expression of TIMP-2 and TSP-1 proteins, with statistical significance, compared to cAMX (*p* < 0.01 and *p* < 0.001, respectively) (Figure 3C,D). cAME demonstrated the highest expression of HGF and TIMP-1 proteins with a significant decrease in HGF protein in cAM compared to cAME (*p* < 0.01) (Figure 3A,B). The highest expression of IL-1RA protein was detected in cAMX (Figure 3E).

### 3.3. cAME and cAMX Demonstrated Corneal Wound Healing Efficacy In Vitro

The concentrations of cAME and cAMX for cell viability and wound healing assays were 2.63 ± 0.12 and 2.54 ± 0.05 mg/mL, respectively. Bacterial identification of cAME and cAMX showed no detection of any bacteria contamination.

#### 3.3.1. Cell Viability Assay

For cAME-treated hCECs, the cell viability of hCECs treated with cAME concentrations of 0.5 mg/mL, 1.0 mg/mL, 1.5 mg/mL and 2.0 mg/mL at 0, 24 and 48 h were analyzed. At 48 h, the percentages of cell viability were 154%, 139%, 130% and 108%, respectively. However, statistical significance was detected only at the 0.5 mg/mL concentration when compared to the beginning hour (*p* < 0.05) (Figure 4A). The cumulative percentages of cell viability at all time points of all cAME concentrations were 123%, 113%, 112% and 97.7%, respectively. The two best concentrations of cAME were at 0.5 mg/mL and 1.0 mg/mL and, therefore, these two concentrations were selected for further study in the wound healing assay.

The cell viability of hCECs treated with cAMX at concentrations of 0.5 mg/mL, 1.0 mg/mL, 1.5 mg/mL and 2.0 mg/mL at 0, 24 and 48 h were analyzed. The cell viability percentages at the end of the study were 88.4%, 93.4%, 89.9% and 64.7%, respectively. A significant decrease in cell viability was observed for cAMX of 2.0 mg/mL concentration at 48 h (*p* < 0.01) (Figure 4B). The cumulative percentages of cell viability at all time points were 93.7%, 99.5%, 93.4% and 82.0%, respectively. Therefore, the best two concentrations of cAMX were at 0.5 mg/mL and 1.0 mg/mL and these two concentrations were also selected for further study in the wound healing assay.

#### 3.3.2. Wound Healing Assay

Photographs of hCECs treated with cAME showed that cells were migrating into wound areas according to time. The wound area of the 1.0 mg/mL cAME-treated group was completely filled at the end of the study (Figure 5A). At the end of the study, percentages of wound closure for the negative control, positive control, cAME 0.5 mg/mL and cAME 1.0 mg/mL were 34.69%, 100.00%, 78.44% and 100.0%, respectively. There were statistically significant differences between both cAME concentrations and the negative control group at every time point (*p* < 0.0001). However, a statistically significant difference between both cAME concentrations was observed at 48 and 72 h post-wounding (*p* < 0.05) (Figure 5B).

For cAMX-treated hCECs, Figure 6A shows that cells were slowly migrating into the wound area over time. However, no group was completely healed at the end of the study except the positive control group. Percentages of wound closure for the negative control, positive control, cAMX 0.5 mg/mL and cAMX 1.0 mg/mL were 34.69%, 100.00%, 64.50% and 66.02%, respectively. There were statistically significant differences between cAMX 1.0 mg/mL and the negative control group at 48 h post-wounding (*p* < 0.05) and between both cAMX concentrations, 0.5 mg/mL and 1.0 mg/mL, and the negative control group at the end of the study (*p* < 0.05 and *p* < 0.01, respectively). However, a statistically significant difference between both cAMX concentrations was not observed at any time points (Figure 6B).

When comparing percentages of wound closure among cAME and cAMX concentrations, cAME at 1.0 mg/mL showed the best concentration in wound closure efficacy at all time points with statistical significance (Figure 7).

## 4. Discussion

To our knowledge, this is the first study to develop topical formulations of cAM, cAME and cAMX. Biological compositions associated with corneal wound healing in cAM, cAME and cAMX were identified by the presence of HGF, TIMP-1, TIMP-2, TSP-1 and IL-1RA at both gene and protein expression levels. In addition, the potencies of cAME and cAMX were also assessed in a corneal wound healing assay in vitro.

The characteristics of cAM in this study included some gross appearance variations, such as size of the membranes and the transparency or vascularization of the membranes. These might be attributed to different breeds, size of bitches and puppies and also their health conditions. In humans, several studies showed that the structure and bioactive compositions of hAM can vary between various donors [24], gestational age [25], region of the membrane and delivery method [26], preservation method [27] and storage duration [28]. As for cAM, a few studies have observed the effect of cryopreservation on the level of EGF [29] and TIMP-1 [30]. We did not focus on these different variation factors in this study; to avoid these variations, similar characteristics of cAM samples were equally divided among each group to obtain overall similar characters as much as possible.

Gene and protein expression results demonstrated the identification of all relevant genes and proteins in cAM, cAME and cAMX. The expression of the HGF gene in cAM was similar to that reported in cAM-derived stem cells [17]. To our knowledge, however, there was no other previous study of the HGF protein presented either directly from cAM itself or its extract. Similarly, the detection of the HGF gene was also reported in hAM [5] and the detection of the HGF protein in hAM [5], hAMH [31] and sterilized powdered hAM [32]. The presence of TIMP-1 gene expression in cAM was similar to the detection in hAM [33]. However, most of the TIMP-1 expression studies were performed at the protein level. From our study, the TIMP-1 protein was identified in cAM similar to a recent study by Dower et al. (2021) [30]. Moreover, the TIMP-1 protein was also detected in hAM [33] and bovine AM homogenate (bAMH) [34]. In the same study, the TIMP-1 protein was not detected in equine AM homogenate (eAMH) [34]. In the case of TIMP-2 expression, the presence of the TIMP-2 gene was identified in cAM similar to that in hAM [33]. As for protein level, the expression of TIMP-2 was also detected in bAMH and eAMH [34]. A similar finding of TSP-1 at gene level in cAM was reported in hAM [33]. At protein level, the TSP-1 protein was also identified in hAM [35], bAM [36] and eAM [37]. However, available data for TSP-1 protein expression in AM topical formulations are still limited. In the case of IL-1RA, its high expression detected in cAME, as compared to cAM, may be related to individual variations. A high expression of IL-1RA was found in maternal and fetal plasma with increasing gestational age [38]. The increased production of IL-1RA was evident in placental membrane with the presence of IL-1β and E. coli LPS [39]. Similar identification of IL-1RA in gene and protein levels were found in hAM [33]. In addition, the presence of the IL-1RA protein was also found in bAM [36] and a topical formulation of human AMEED [40].

In the current study, variations in gene and protein expression in cAM, cAME and cAMX were observed. A reason for unpredictable correlation between gene and protein expressions in this study could be due to the individual characteristics of the samples and differences in sample preparation and processing methods. An extraction method by pulverization was reported to achieve more extractable factors compared to other AME preparation methods such as homogenization [41]. This might be because cellular destruction from liquid nitrogen causes damage to the cell wall of the membrane and subsequently causes release of proteins [24,41]. In addition, the lyophilization process causes significant reduction in total protein and growth factor concentrations when compared to the cryopreservation method [42]. However, there were several studies showing that correlation between mRNA and protein levels was not sufficient enough to predict protein expression levels from quantitative mRNA data [43]. This finding points towards the complexity and diversity of regulatory mechanisms responsible for the observed differences, which are post-transcriptional or post-translational parameters, noise and experimental error [44].

In vitro studies of cell viability and wound healing assays were conducted to assess the possibility of cAME and cAMX usage as an eye drop for clinical applications. According to the protein expression study, both cAME and cAMX formulations still contained bioactive molecules that can increase cell viability and promote cell migration and cell growth close to the defective area in the corneal wound healing model.

Cell viability assays of hCECs treated with cAME and cAMX demonstrated different percentages of cell viability. Several factors may play a role in the higher than 100% cell viability in most of cAME and some of cAMX concentrations, such as an increasing metabolic activity of the cells without cell proliferation [45], true cell proliferation [46] or interference of assay from proteins and inhibitors [46]. The decrease in proliferative effect of the cell viability assay in a concentration-dependent manner might be related to several possible factors. Transforming growth factor (TGF) -β showed some inhibitory effects on cell proliferation through the TGF-β signaling pathway [47,48]. Different concentrations of the extracts added to KSFM might alter the osmotic pressure of the cell microenvironment. It then may have an influence on the physiological conditions of cells [49]. Similarly, hyperosmotic conditions, such as dry eye or topical administration of hyperosmotic solution, induce shrinkage of corneal epithelial cells that leads to cell death [49,50]. Additionally, the proportion of the lower volume of KSFM to the higher concentration of the extracts might result in a limited amount of essential nutrients within cells, which might eventually lead to inhibited cell viability [51,52]. In the current study, cAMX at a high concentration of 2.0 mg/mL showed a statistically significant decrease in cell viability at 48 h when compared to cells at the beginning of the study. A similar study was conducted on the use of 0.1–2.0 mg/mL AMEED concentration in limbal stem cell proliferation. The result showed that AMEED at 1.0 mg/mL concentration demonstrated the best growth of cells, while 2.0 mg/mL concentration inhibited cell growth [40]. According to our results, concentrations of 0.5 mg/mL and 1.0 mg/mL of both cAME and cAMX appeared to be the best two concentrations among all those studied, by giving a better cumulative percentage of cell viability.

For the wound healing assay, both cAME and cAMX formulations had corneal epithelial wound healing efficacy. However, the degree of healing varied between these two formulations and the two concentrations. For cAME, wound closure percentages of both concentrations were significantly increased when compared to the negative control group at all time points. In addition, a concentration of 1.0 mg/mL was significantly greater in wound healing percentage than a concentration of 0.5 mg/mL at 48 and 72 h. This is similar to a study in humans, where hCECs treated with AME healed faster after mechanical injury, suggesting a potential benefit in acute corneal injuries [53]. In the case of cAMX, the wound closure percentage of 1.0 mg/mL concentration was significantly increased when compared to the negative control group at 48 h. At the end of the study, both cAMX concentrations were significantly greater in healing ability than the negative control. The healing of cAMX, on the other hand, was lower than that of cAME. According to the results from protein expression in this study, this might relate to significantly lower expression of HGF protein in cAMX compared to cAME. Here, the lyophilization process might cause the differences in proteins or levels of growth factors between these two formulations [42]. The limitations of our study are that, firstly, preserved cAM in RNALater™ solution was used as a control instead of fresh cAM due to limitations in sample collection and the processing method. However, several studies confirmed that RNALater™ could preserve the quality and quantity of mRNA and proteins without significant difference from fresh and snap-frozen tissues [54]. Second, hCECs were used in the in vitro study instead of canine corneal epithelial cells (cCECs) due to the limited lifespan of primary cCECs and the commercial unavailability of immortalized cCECs. Similar mucin expression was detected between human and canine species, which might suggest their similarity in CEC function [55]. Therefore, we anticipate that the use of immortalized hCECs in our study will provide a preliminary result in vitro that may lead to further in vivo study.

## 5. Conclusions

From our study, cAM can be generated into cAME and cAMX formulations for topical use. These formulations contain genes and proteins (HGF, TIMP-1, TIMP-2, TSP-1 and IL-1RA) similar to those reported either in cAM itself or in amniotic membrane from other species. The increased expressions of HGF, TIMP-1 and IL-1RA at both gene and protein levels in cAME might indicate the beneficial properties of cAME in corneal wound healing. Although most protein levels in cAMX were decreased compared to cAME due to the lyophilization process, both cAME and cAMX formulations accelerated the healing of wound defects better than the negative controls. cAME at 1.0 mg/mL concentration appeared to be superior in corneal wound healing efficacy in vitro. Clinical investigation of cAME and cAMX treatments should be conducted to assess the clinical outcomes of corneal wound healing. In addition, the effect of storage duration on the composition and level of growth factors should also be further investigated.

## Figures and Tables

**Figure 1 vetsci-09-00227-f001:**
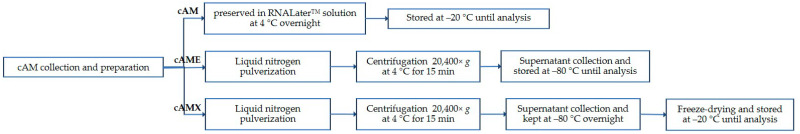
Schematic diagram showing canine amniotic membrane preparation of each group.

**Figure 2 vetsci-09-00227-f002:**
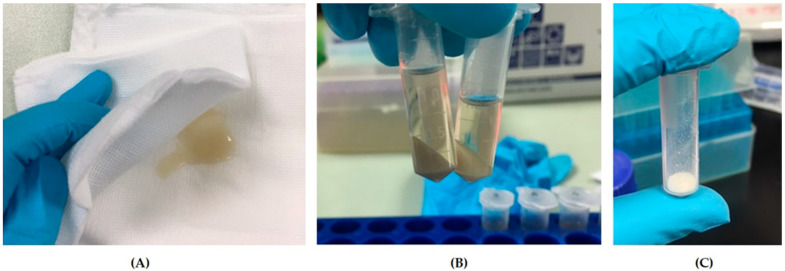
Photographs demonstrating characteristics of each cAM preparations: (**A**) canine amniotic membrane (cAM); (**B**) canine amniotic membrane extract (cAME); and (**C**) lyophilized canine amniotic membrane extract (cAMX).

**Figure 3 vetsci-09-00227-f003:**
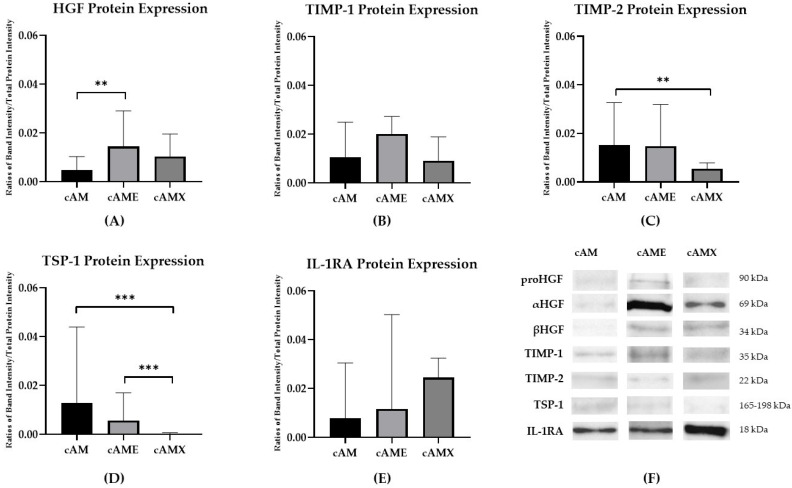
Bar charts showing ratios of band intensities of relevant proteins normalized by total protein intensity: (**A**) HGF protein; (**B**) TIMP-1 protein; (**C**) TIMP-2 protein; (**D**) TSP-1 protein; and (**E**) IL-1RA protein. (**F**) Western blot analysis representing relevant proteins expression in cAM, cAME and cAMX (The original western blot was shown in Appendix A). Double asterisks (**) and triple asterisks (***) indicate significance levels of *p* < 0.01 and *p* < 0.001, respectively.

**Figure 4 vetsci-09-00227-f004:**
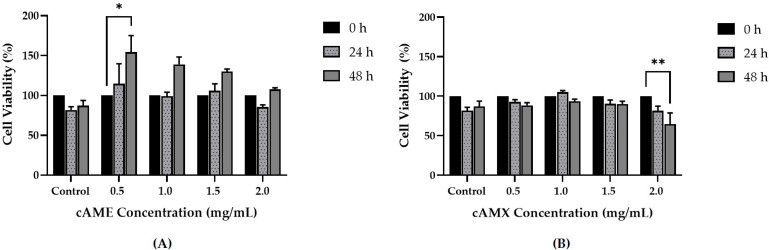
Bar graphs showing the percentages of cell viability at various concentrations and time points. (**A**) Cell viability of hCECs treated with cAME. (**B**) Cell viability of hCECs treated with cAMX. Asterisk (*) and double asterisks (**) indicate significance levels of *p* < 0.05 and *p* < 0.01, respectively.

**Figure 5 vetsci-09-00227-f005:**
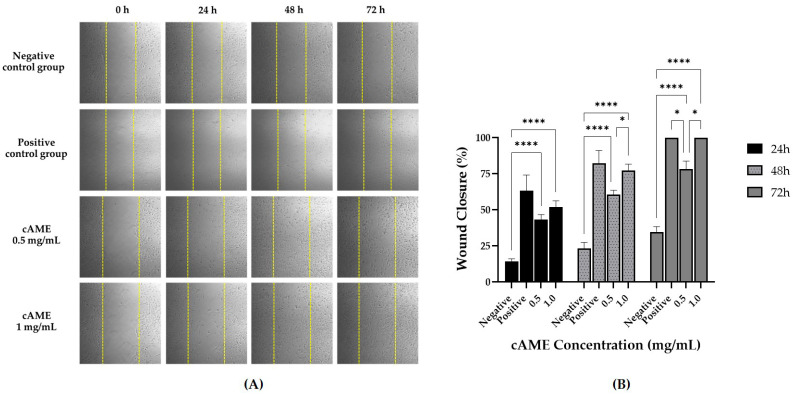
Wound healing assay of cAME. (**A**) Pictures showing migration of hCECs into the wound area (5× magnification). The yellow dotted line represents wound edge measuring at the beginning of the study. (**B**) Bar graph showing comparisons of wound closure percentages among groups at 0, 24, 48 and 72 h post-wounding. Asterisk (*) and quadruple asterisks (****) show significance levels of *p* < 0.05 and *p* < 0.0001, respectively.

**Figure 6 vetsci-09-00227-f006:**
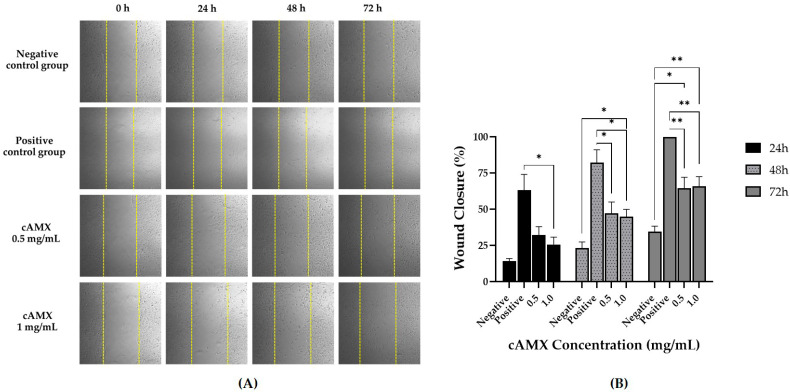
Wound healing assay of cAMX. (**A**) Pictures showing migration of hCECs into the wound area (5× magnification). The yellow dotted line represents wound edge measuring at the beginning of the study. (**B**) Bar graph showing comparison of wound closure percentages among groups at 0, 24, 48 and 72 h post-wounding. Asterisk (*) and double asterisks (**) show significance levels of *p* < 0.05 and *p* < 0.01, respectively.

**Figure 7 vetsci-09-00227-f007:**
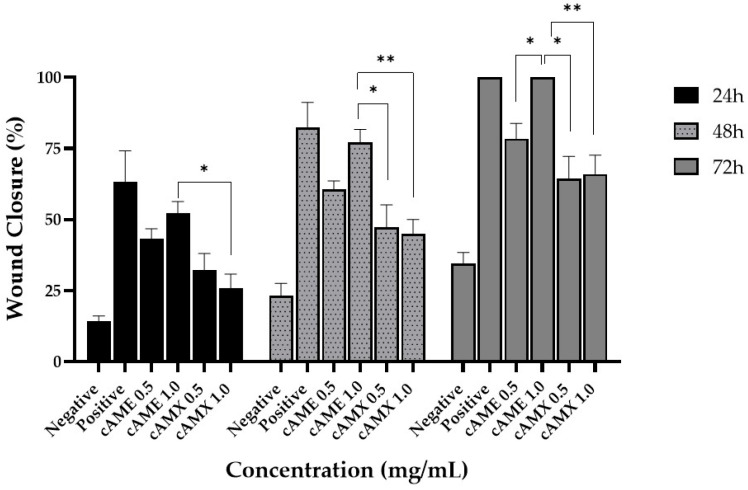
Bar graph showing a comparison of wound closure percentages among all experiments at 24, 48 and 72 h post-wounding. Asterisk (*) and double asterisks (**) indicate significance levels of *p* < 0.05 and *p* < 0. 01, respectively.

**Table 1 vetsci-09-00227-t001:** Primer sequences and amplicon sizes of relevant genes.

Genes	Primer Sequences	Accession Number	Amplicon Size (bp)
HGF	Fw: 5′-AAAGGAGATGAGAAACGCAAACAG-3′ Rv: 5′-GGCCTAGCAAGCTTCAGTAATACC-3′	AB090353.1	92
TIMP-1	Fw: 5′-CTCACCAGAGAACCCACCAT-3′ Rv: 5′-CCTGATGACGATTTGGGAGT-3′	AB016817.1	147
TIMP-2	Fw: 5′-AGCAGCACCCAGAAGAAGAG-3′ Rv: 5′-GTCCATCCAGAGGCACTCAT-3′	NM_001003082.1	120
TSP-1	Fw: 5′-CGCGATGGCAGCTGGAAATGTG-3′ Rv: 5′-GGGCAGGGCAGGCAGTTGTAGC-3	XM_544610.6	167
IL-1RA	Fw:5′-GAAGAGACCTTGCAGGATGC-3′ Rv: 5′-GACGGGCACCACATCTAACT-3′	AY026462.2	142
GAPDH	Fw: 5′-TGTCCCCACCCCCAATGTATC-3′ Rv: 5′-CTCCGATGCCTGCTTCACTACCTT-3′	AB038240.1	100

Fw: Forward, Rv:Reverse, bp: base pair.

**Table 2 vetsci-09-00227-t002:** Relative expression of relevant genes in fold change.

Genes	cAME (cAM)		cAMX (cAM)		cAMX (cAME)	
	Fold Change	*p* Value	Fold Change	*p* Value	Fold Change	*p* Value
HGF	1.561 (0.291–12.123)	0.455	0.478 (0.146–1.613)	0.048 *^,b^	0.306 (0.031–2.071)	0.045 *^,c^
TIMP-1	2.162 (0.737–8.165)	0.024 *^,a^	1.669 (0.824–3.116)	0.008 **^,b^	0.772 (0.195–2.143)	0.404
TIMP-2	0.670 (0.233–2.143)	0.218	0.730 (0.274–1.705)	0.237	1.089 (0.330–3.031)	0.788
TSP-1	0.941 (0.198–8.224)	0.900	0.422 (0.174–1.042)	0.001 **^,b^	0.449 (0.058–2.411)	0.129
IL-1RA	10.061 (1.133–125.353)	0.000 ***^,a^	2.152 (0.914–5.314)	0.009 **^,b^	0.214 (0.017–1.777)	0.012 *^,c^

Control group is shown in parentheses in the table headings. The fold change with standard error (S.E.) is shown in parentheses in the table contents. Asterisk (*), double asterisks (**) and triple asterisks (***) indicate statistical significance levels at *p* < 0.05, *p* < 0.01 and *p* < 0.001, respectively. ^a^, ^b^ and ^c^ show differences between cAM and cAME, cAM and cAMX, and cAME and cAMX, respectively.

## Data Availability

The data presented in this study are available on request from the corresponding author.

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
