# Peer review of "Biological Compositions of Canine Amniotic Membrane and Its Extracts and the Investigation of Corneal Wound Healing Efficacy In Vitro"

_vetsci, 2022, doi:10.3390/vetsci9050227_

Round 1
Reviewer 1 Report
The paper is well written and provides important information concerning topical canine amniotic membrane formulations on corneal wound healing.
Author Response
Dear Reviewer 1,
We truly appreciate your comments.
Best regards,
Authors
Reviewer 2 Report
The authors revealed that canine amniotic membrane and its etracts contain bioactive molecules related to corneal wound healing and have wound healing properties.
This manuscript is interesting, however, there are some questions.
- Why did the authors use human cell line? I think that canine cells should be used.
- In figure 2, why dose "cell viability" be over 100%? I think it is not "cell viability" but "cell proliferation ratio".
- Why dose cAME promote cell proliferation in concentration-dependent manner in Figure 3 although it inhibits cell proliferation in concentration-dependent manner in Figure 2?
Author Response
Dear Reviewer 2,
Please find the attached file for the revision 1 of the manuscript 1692932.
Best regards,
Authors

Reviewer 3 Report
- Materials and Methods
2.2. Canine amniotic membrane preparation
Please specify how many cAM were used in total. 120 cAM were used. 84 cAM were used for gene and protein expression. 36 cAM were used for wound healing assay.
It is difficult to understand how to make cAME from cAM. Similarly, it is difficult to understand how to make cAMX from cAME. Please show this in a schematic diagram in addition to the text.
2.3. Gene expression study by quantitative reverse transcription polymerase chain reaction
2.3.1. RNA isolation
Indicate the amount of RNA isolation buffer used for cAM, cAME and cAMX, respectively.
- Results
3.1. cAME and cAMX can be generated from cAM
It is difficult to understand what each sample IS. Is cAM a membrane, cAME a liquid, and cAMX a powder? Please provide a typical picture of each for better understanding.
3.2. Biological compositions associated with corneal wound healing were identified in cAM, cAME and cAMX both gene and protein expression levels
You should measure EGF. EGF is one of the most important growth factors in wound healing. You have to explain why you did not measure EGF.
3.2. Biological compositions associated with corneal wound healing were identified in cAM, 254 cAME and cAMX both gene and protein expression levels
Table 2. Relative expression of relevant genes in fold change.
Is this the mean value of the samples derived from 14 cAMs? If so, please state the standard deviation.
IL-1RA in cAME is a major anomaly. Were the values of all 14 individuals high? Was there a large variation among individuals? The reason for this large abnormality must be explained.
3.3.1. Cell viability assay
As shown in the line 201~203, the hCECs were resuspended with 100 μ L of KSFM containing various concentration of cAME and cAMX (0.5, 1, 1.5, 2 mg/mL) and seeded at a concentration of 1 x 104 cells/well 202 into 96-well plates. And in the line 278~279, the concentrations of cAME and cAMX for cell viability and wound healing assay were 2.63 ± 0.12 and 2.54 ± 0.05 mg/mL, respectively. The 2mg/ml and 0.5mg/ml media have different concentrations of KSFM in them. This makes it impossible to legitimately compare different concentrations. If it is not possible to redo the experiment, the effect on the results must be clearly stated in the DISCUSSION.
3.3.2. Wound healing assay
In Figures 3, 4 and 5, both positive and negative control must be presented.
In Figure 3 and 4, the position of the yellow wavy line is different in the control group. The position of the yellow wavy line must be unified.
Author Response
Dear Reviewer 3,
Please find the attached file for the revision 1 of the manuscript 1692932.
Best regards,
Authors

Reviewer 4 Report
Good paper- congratulations.
You wrote- Canine amniotic membrane (cAM) was collected from healthy puppies delivered by cesarean section. Important info are the age and the general condition of puppies' mothers. Also the species should be the same.
Could be more precise in few details of 2.2. Canine amniotic membrane preparation. Eighty-four cAMs were equally divided into 3 groups according to their preparation methods. Group 1, cAM (n = 28), was immersed in RNAlater™ solution (Thermo Fisher 90 Scientific, Waltham, MA, USA), stored at 4 °C overnight, then kept at – 20 °C until analysis. What analysis? What exactly did you do with cAM before immersing in RNAlater™ solution? I suspect that cAM were somehow damaged to lyse the cells.
Group 2, cAM extract (cAME, n = 28), was pulverized using liquid nitrogen until it became fine particles. How many repetitions (freeze- thawing) were needed to get fine particles?
After thawing, the suspension was collected and centrifuged at 20,400 x g for 15 min at 4 °C. The supernatant was collected and stored at – 80 °C until the
experiment. How long was supernatant stored?
Group 3, lyophilized cAME (cAMX, n = 28), was prepared using a similar process to cAME. The supernatant was kept at – 80 °C overnight, then transferred to a freeze dry system (Labconco LYPH·LOCK®4.5, Kansas City, MO, USA). After processing, the tubes containing cAME powder were kept at – 20 °C until analysis.
Why -20 not -80C storage temp?
Could you describe more precisely the freeze-drying process? media used? the freezing program used? how many degress per minute?
These are important details if others are to repeat the experiments. the production of curing material.
In disussion you wrote "the lyophilization process causes significant 381 reduction in total protein and growth factor concentrations when compared to the cryo-382 preservation method [41]" If so, why did you choose this method?
Could you propose what factors may be responsible for observation that cAMX at a high concentration of 2.0 mg/mL showed a statistically significant decrease of cell viability at 48 h when compared to cell viability at the beginning of the study.
Generally I think that the paper brings new original data important for clinical practice.
Author Response
Dear Reviewer 4,
Please find the attached file for the revision 1 of the manuscript 1692932.
Best regards,
Authors
